# Diagnostic Accuracy of Magnetic Resonance Imaging for Sagittal Cervical Spine Alignment: A Retrospective Cohort Study

**DOI:** 10.3390/ijerph182413033

**Published:** 2021-12-10

**Authors:** Chahyun Oh, Chan Noh, Jieun Lee, Sangmin Lee, Boohwi Hong, Youngkwon Ko, Chaeseong Lim, Sun Yeul Lee, Yoon-Hee Kim

**Affiliations:** 1Department of Anesthesiology and Pain Medicine, Chungnam National University Hospital, Daejeon 35015, Korea; ohchahyun@gmail.com (C.O.); jyfchrh@naver.com (C.N.); peppermintje@naver.com (J.L.); lkb333@naver.com (S.L.); koho0127@gmail.com (B.H.); annn8432@gmail.com (Y.K.); limtwo2@gmail.com (C.L.); 2Department of Anesthesiology and Pain Medicine, College of Medicine, Chungnam National University, Daejeon 35015, Korea

**Keywords:** cervical spine, magnetic resonance imaging, radiography, spinal curvatures

## Abstract

(1) Background: Although radiography performed on the subject in an upright position is considered the standard method for assessing sagittal cervical alignment, it is frequently determined, or reported, based on MRI performed on the subject in a supine position. (2) Methods: Cervical alignment observed in both imaging modalities was assessed using four methods: the C2-7 Cobb angle, the absolute rotation angle (ARA), Borden’s method, and the sagittal vertical axis (SVA). Cervical alignment was determined (lordosis, kyphosis, and straight) based on radiography. Then, the diagnostic cut-off values for the MRI images and their corresponding diagnostic accuracies were assessed. (3) Results: The analysis included 142 outpatients. The determined diagnostic cut-off values for lordosis, using three measurements (Cobb angle, ARA, and Borden’s method), were −8.5°, −12.5°, and 3.5 mm, respectively, and the cut-off values for kyphosis were −4.5°, 0.5°, and −1.5 mm, respectively. The cut-off value for SVA > 40 mm was 19.5 mm. The Cobb angle, ARA, and Borden’s method, on MRI, showed high negative predictive values for determining kyphosis. The SVA on MRI measurements also showed high negative predictive values for determining >40 mm. (4) Conclusions: MRI measurements may be predictive of cervical alignment, especially for the exclusion of kyphosis and SVA > 40 mm. However, caution is needed in the other determinations using MRI, as their accuracies are limited.

## 1. Introduction

Cervical malalignment has been associated with the development of cervical pathologies [1,2] and reduced health-related quality of life [3]. A recent large-scale retrospective study [4] revealed that there is an increasing trend of cervical malalignment. Assessing cervical spine alignment, therefore, has become increasingly important.

Although radiography with the patient in an upright position is considered the standard method for assessing sagittal cervical alignment [5], cervical alignment is frequently determined, or reported, based on magnetic resonance imaging (MRI) or computed tomography (CT), performed on subjects in a supine position [6,7,8]. However, the difference in the patient positioning and the use of a head stabilizer during MRI scanning may affect cervical alignment. Therefore, direct comparisons between these two modalities may be misleading. In addition, the diagnostic cut-off values and accuracy of these determinations remain unclear. The present study, therefore, retrospectively assessed the diagnostic accuracy of MRI in the assessment of cervical spine alignment, by comparing the MRI images with those from upright cervical radiography as a reference, using four well-known measurements [5,9].

## 2. Materials and Methods

### 2.1. Study Design and Population

The protocol of this retrospective study was approved by the Institutional Review Board of Chungnam National University Hospital (CNUH 2020-08-092). This study included outpatients who underwent cervical radiography (upright lateral images) and MRI (supine sagittal images) during the year 2019 at Chungnam National University Hospital, a tertiary teaching hospital. Patients were excluded if they underwent post-operative evaluations or had experienced extensive trauma. Patients were also excluded if they were aged >70 years; had anatomical abnormalities that could significantly distort the alignment (e.g., bone or soft tissue tumor); had undergone the two examinations (radiography and MRI) at intervals > 3 months; and had images in which the alignment could not be assessed, including those with ill-defined bony margins and those in which images of the seventh cervical vertebra were blocked by the shoulder. Patient characteristics, including age, sex, height, weight, body mass index (BMI), and the date of examination, were recorded. This manuscript adheres to the applicable STARD (Standards for Reporting of Diagnostic Accuracy Studies) guidelines [10].

### 2.2. Measurements

Patients underwent upright lateral cervical radiography focusing on the fourth cervical vertebra at a distance of 180 cm. Patients also underwent cervical MRI in a supine position using a standard head stabilizer (Appendix A). Details regarding the MRI machines used in 2019 are summarized in Appendix A. The MRI assessments were based on T2-weighted sagittal images crossing the center of the cervical spine. All parameters were measured by an experienced pain clinic physician (C.N., who had completed a 2-year fellowship and 3 years as a clinical professor in a tertiary hospital) using the basic tools of the Picture Archiving and Communicating System (PACS; Maroview, Marosis, Korea). Cervical alignment on both upright cervical radiography (standard lateral view) and MRI (Figure 1) was assessed using four methods [5,9]:

C2-7 Cobb angle: two lines were drawn, one each parallel to the inferior endplate of the second (C2) and seventh (C7) cervical vertebrae, and the angle between these two lines was measured.

C2-7 absolute rotation angle (ARA): two lines were drawn, one each parallel to the posterior margin of the vertebral bodies of C2 and C7, and the angle between these two lines was measured.

Borden’s method: A straight line was drawn connecting the posterior edge of the odontoid process and the posterior inferior tip of the vertebral body of C7. The longest vertical distance from this line to the imaginary line connecting the posterior margins of the C2 to C7 vertebral bodies was measured.

C2-7 sagittal vertical axis (SVA): A vertical line was drawn from the center of the vertebral body of C2 to the ground. The vertical distance from this line to the posterior superior aspect of the C7 vertebral body was measured.

The cut-off values for determining loss of lordosis and kyphosis were defined as follows:

Lordosis < −10°; straight −10° to 0°; kyphosis > 0°, for the C2-7 Cobb angle and the ARA [11,12].

Lordosis > 7 mm; straight 7 to 0 mm; kyphosis < 0 mm, for Borden’s method [9].

The measurements of C2-7 SVA were stratified by the cut-off value of 40 mm [13].

### 2.3. Statistical Analysis

The sample size was based on the data available from January to December 2019. No statistical power calculation was performed before the study. All statistical analyses were performed using R software, version 4.0.2 (R Project for Statistical Computing, Vienna, Austria). Continuous variables were analyzed by the independent t-test (mean ± SD), or the Kruskal–Wallis test (median [IQR]), depending on the results of Shapiro–Wilk tests.

Because the parameters were measured in different positions, inter-class correlation coefficients (ICCs), Spearman correlation coefficients specifically, were calculated for each paired measurement. The cut-off values for each measurement on MRI, which maximize the Youden index (sensitivity + specificity − 1), were determined by receiver operating curves (ROCs) based on the results of radiography. The area under the curve (AUC) for the ROC curves was compared using the Delong test [14]. Values of *p* < 0.05 were defined as statistically significant.

As an exploratory analysis, correlations between age and the radiographic measurements were assessed to explore the influence of age on cervical alignment.

## 3. Results

During the study period, 248 patients underwent both upright lateral cervical radiography and cervical MRI in the supine position. Of these, 106 were excluded, including 48 patients aged >70 years, 32 who had undergone previous c-spine surgery, 3 with a poor image quality, 15 with a masked C7, 1 with a c-spine fracture, and 7 who underwent the two examinations at intervals >3 months. Thus, the analysis included 142 outpatients from 5 departments: orthopedic surgery, rehabilitation medicine, neurosurgery, neurology, and anesthesiology and pain medicine (Figure 2). The characteristics of these patients are summarized in Table 1. Height was not recorded in 72 patients, and weight was not recorded in 67. Values for C2-7 SVA on radiography were significantly lower in female patients. Other parameters showed no significant difference by sex.

All paired measurements showed a significant correlation (C2-7 Cobb angle: 0.527, C2-7 ARA: 0.488, Borden’s method: 0.481, C2-7 SVA: 0.535; *p* < 0.001 each) (Figure 3). There were significant correlations between the radiographic measurements and the difference between the two paired measurements (radiographic—MRI) (C2-7 Cobb angle: 0.405, C2-7 ARA: 0.404, Borden’s method: 0.533, C2-7 SVA: 0.837; *p* < 0.001 each) (Figure 4). Patients showing more lordotic curvatures on radiography showed a greater kyphotic change on MRI. In addition, patients with greater C2-7 SVA values on radiography tended to show a larger decrease in C2-7 SVA values on MRI.

The ROCs and AUCs of these four measurements on MRI are shown in Figure 5 and Table 2. The determined diagnostic cut-off values of MRI for kyphosis using three measurements (i.e., C2-7 Cobb angle, C2-7 ARA, and Borden’s method) were −4.5°, 0.5°, and −1.5 mm, respectively. The cut-off value of MRI for C2-7 SVA > 40 mm was 19.5 mm. The determined diagnostic cut-off values and their corresponding diagnostic accuracies are summarized in Table 3. There were no significant differences in AUCs between pairs of the three measurements for lordosis and kyphosis.

Based on the newly determined cut-off values, the C2-7 Cobb angle, the C2-7 ARA, and Borden’s method showed high negative predictive values for determining kyphosis based on MRI measurement (0.977, 0.965, and 0.939, respectively). C2-7 SVA also showed high negative predictive value for determining C2-7 SVA > 40 mm based on MRI measurement (0.992). Cervical alignments, as determined by radiography and MRI, are cross-tabulated in Appendix A.

The result of the exploratory analysis showed no significant correlation between age and the four measurements on radiography (Appendix A).

## 4. Discussion

This study showed that MRI measurements of cervical alignment in the supine position correlated significantly with upright radiographic measurements. Although the relationship between these two imaging modalities can allow cervical alignment to be predicted based on MRI measurements, these relationships are not perfectly linear, nor do they have regression coefficients near 1.0. Thus, MRI measurements cannot completely replace radiographic measurements for the assessment of cervical alignment, nor can the same cut-off values be applied.

Based on the result of the current study, kyphosis and C2-7 SVA > 40 mm can be reliably ruled out by measurements on MRI. The evaluation of their negative predictive values showed excellent accuracy for all four methods. Lack of kyphosis on MRI was highly indicative of lack of kyphosis on radiography. In contrast, ‘loss of lordosis’ on MRI may not indicate a similar finding on radiography. The likelihood that ‘loss of lordosis’ is correct based on MRI measurements using the C2-7 ARA was only 56.7%. If ‘loss of lordosis’ is observed on MRI, it can be a result from kyphotic change due to positional change from upright to supine. Therefore, determination of cervical alignment based on MRI is limited. Distinct cut-off values, as suggested in this study, should be used to minimize the loss of diagnostic accuracy.

A previous prospective observational study also found similar correlations between radiography and MRI measurements of cervical alignment, with correlation coefficients for the C2-7 Cobb angle and the ARA being 0.55 and 0.48, respectively [15]. That study, however, used different cut-off values for determining cervical alignment. The evaluation method was similar to Borden’s method, except that the alignment category depended on whether the posterior margin of the vertebral body crossed or was on the line connecting the lower back ends of C2 and C7. Although that study also reported ‘neutral’ alignment, identical to the ‘straight’ alignment in the current study, the absence of a detailed cut-off value introduced some degree of ambiguity. The cut-off values derived from the current study can be used to determine a broader range of neutral alignment. This broader intermediate zone (i.e., straight alignment) may enable much more meaningful classification, distinguishing between significant and intermediate degrees of malalignment.

Although symptoms, rather than anatomic changes in imaging results, are considered important, measured cervical alignment itself has clinical value in the pathophysiology and prognosis of the cervical spine. Cervical malalignment was shown to have negative effects on degenerative changes [2] and disc herniation [1] of the cervical spine. Cervical malalignment has also been associated with increased neck disability and reduced health-related quality of life (HRQOL) [3,13,16]. Moreover, according to a recent study [4], there was an increasing trend of loss of lordotic curvature across time from 2006 to 2018, especially in younger patients. The authors of the study pointed out that the predominant use of the internet and handheld devices may be associated with this increase in cervical malalignment in younger patients. Many handheld device users flex their neck while viewing the device [17,18,19,20], which can cause harm to the cervical spine [21,22]. As this concerning prevalence of abnormal curvature is expected to continue, the clinical importance of the assessment of cervical alignment is growing.

The results of the exploratory analysis in this study show that no significant correlation between age and cervical curvature exists. Nonetheless, this may require careful interpretation. In contrast to the current study, a previous study [23] showed a clear correlation between age and lordosis (increased lordosis with increased age). This may be due to the differences between the cohorts involved in these studies. In the previous study, most of the patients were females (70.4%) and of a relatively younger (mean age 48.1) and evenly distributed age. On the contrary, the cohort in the current study had a similar number of male and female patients, with an unevenly distributed age (tilted toward older ages). There were only 23 (about 16%) patients that were aged under 40 in this study. Detailed analysis of this issue is beyond the scope of the current study and warrants further study.

Replacing a simple and inexpensive standard method (radiography) with a non-standard and much more expensive method (MRI) may be counterintuitive. However, there are several situations in which this replacement can be justified. First, standard radiography may not always be possible. For example, in the current study, cervical alignment could not be assessed radiographically in 18 patients due to poor image quality or an obscured C7 vertebral body. In addition, physical or medical constraints may prevent patients from being in an upright position. Second, retrospective analysis using supine MRI or CT images may be the only option [6,7,8]. Other than these situations, a clinician should at least keep in mind that upright radiography cannot be completely replaced by supine MRI, and that the assessment of cervical alignment based on MRI measurements may be inaccurate.

This study had several limitations. First, the study subjects consisted of patients who visited a tertiary hospital. Generalizability to other settings, including primary care and emergency settings, requires further validation. However, because the study subjects consisted of outpatients and excluded those who had undergone previous surgery or had extremely abnormal anatomies, the subjects of this study likely represent typical outpatients who undergo cervical spine imaging. Second, this study did not report detailed clinical information, such as the patients’ chief complaints, which may have affected the alignment. Third, as there were many missing data on patients’ weight and height, detailed analysis regarding the influence of these factors on cervical alignment was not available.

## 5. Conclusions

In conclusion, this study shows that MRI measurements of cervical alignment correlated significantly with upright radiographic measurements. MRI measurements may, therefore, be predictive of cervical alignment, especially for the exclusion of kyphosis and C2-7 SVA > 40 mm. However, caution is needed in the other determinations using MRI, as their accuracies are limited.

## Figures and Tables

**Figure 1 ijerph-18-13033-f001:**
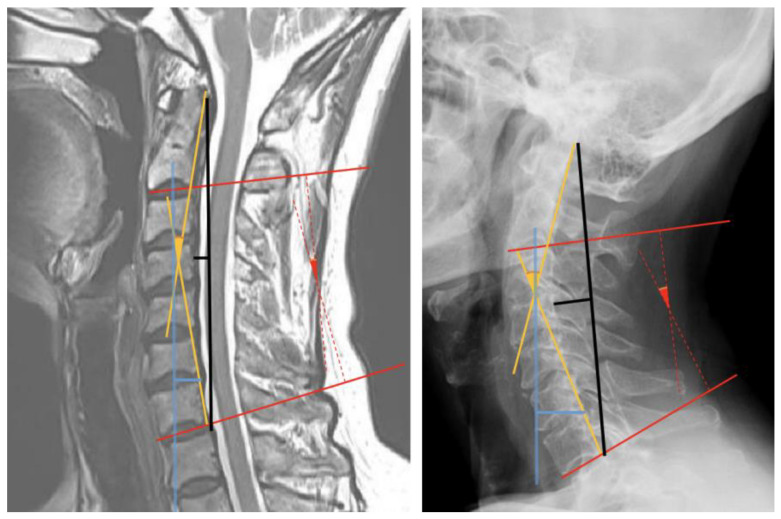
Measurements of cervical alignment shown on a magnetic resonance image (**left image**) and a radiograph (**right image**). The solid and dotted red lines indicate measurements of the C2-7 Cobb angles. The yellow solid lines indicate measurements of the C2-7 absolute rotational angle. The black solid lines indicate measurements using Borden’s method. The blue solid lines indicate measurements of the C2-7 sagittal vertical axis.

**Figure 2 ijerph-18-13033-f002:**
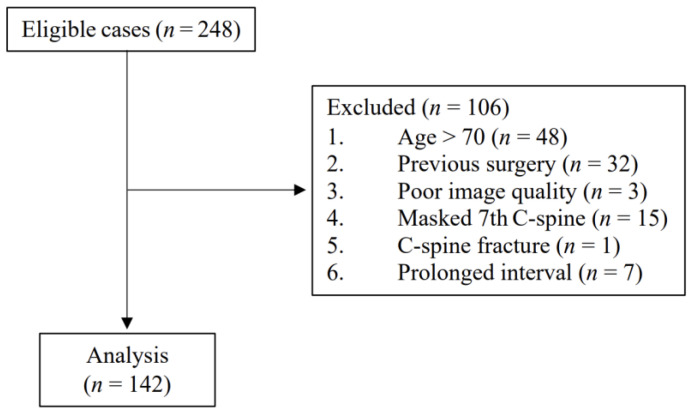
Patient flow diagram.

**Figure 3 ijerph-18-13033-f003:**
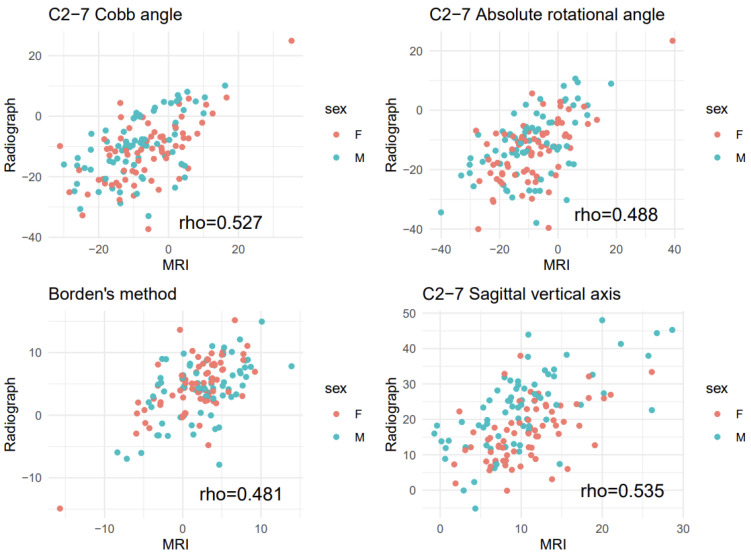
Scatter plot of radiographic and magnetic resonance imaging (MRI) measurements. Rho indicates Spearman correlation coefficient (without stratification by sex).

**Figure 4 ijerph-18-13033-f004:**
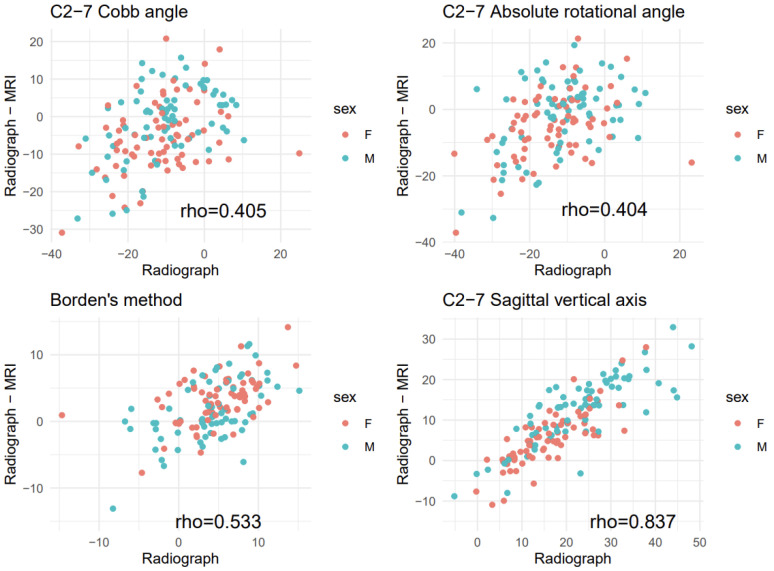
Scatter plots for radiographic measurements versus difference between the two paired measurements (radiographic—magnetic resonance imaging (MRI)). Rho indicates Spearman correlation coefficient (without stratification by sex).

**Figure 5 ijerph-18-13033-f005:**
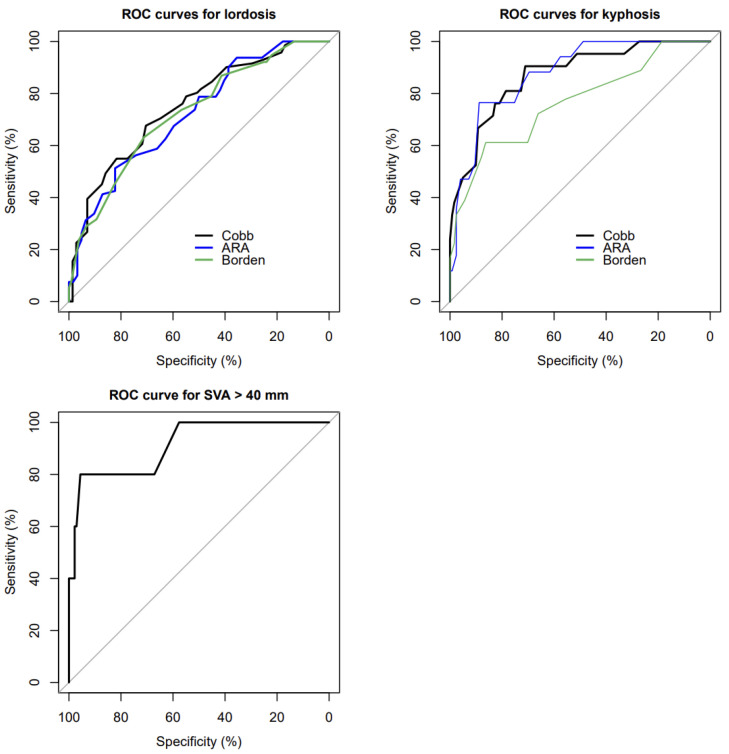
ROC curves for lordosis, kyphosis, and C2-7 SVA > 40 mm. ROC: receiver operating curve, Cobb: C2-7 Cobb angle, ARA: C2-7 absolute rotational angle, Borden: Borden’s method, SVA: C2-7 sagittal vertical axis.

**Table 1 ijerph-18-13033-t001:** Patient characteristics.

Characteristics	Total	Female	Male	*p*
	(*n* = 142)	(*n* = 70)	(*n* = 72)	
Age (y)	52.5 (5.0, 62.0)	54.5 (47.0, 63.0)	51.5 (43.5, 61.0)	0.18
Height (cm)	162.9 ± 9.0	156.3 ± 6.0	169.5 ± 6.4	<0.001
Weight (kg)	63.8 (55.4, 71.0)	58.9 ± 10.3	70.6 ± 12.5	<0.001
BMI (kg/m^2^)	24.3 ± 3.7	24.1 ± 3.7	24.5 ± 3.8	0.636
Interval (day) *	4.0 (0.0, 10.0)	3.0 (0.0, 9.0)	6.0 (0.0, 11.0)	0.180
Radiography				
C2-7 Cobb angle (°)	−10.8 ± 10.3	−11.6 ± 10.3	−10.0 ± 10.2	0.364
C2-7 ARA (°)	−12.6 ± 10.8	−13.8 ± 10.7	−11.5 ± 10.8	0.204
Borden’s method (mm)	5.0 (2.0, 8.0)	5.0 (2.0, 8.0)	4.0 (2.5, 7.5)	0.502
C2-7 SVA (mm)	19.4 ± 10.1	16.0 ± 8.1	22.7 ± 10.8	<0.001
MRI				
C2-7 Cobb angle (°)	−7.4 ± 10.8	−6.1 ± 11.3	−8.6 ± 10.2	0.173
C2-7 ARA (°)	−10.0 (−17.0, −2.0)	−8.9 ± 11.2	−9.8 ± 11.5	0.666
Borden’s method (mm)	3.0 (0.0, 4.0)	2.5 (0.0, 4.0)	3.0 (0.0, 5.0)	0.286
C2-7 SVA (mm)	10.0 (7.0, 13.0)	10.0 (7.0, 13.0)	10.0 (6.0, 12.5)	0.486

Values are mean ± SD or median (IQR). * The interval between radiography and magnetic resonance imaging (MRI). Abbreviations: BMI, body mass index; ARA, absolute rotational angle; SVA, sagittal vertical axis.

**Table 2 ijerph-18-13033-t002:** Area under the curve of the four measurements.

Alignment	Measurements	AUC	SE	95% CI	*p*
Lordosis	C2-7 Cobb angle	0.749	0.041	0.669 to 0.828	<0.001
	C2-7 ARA	0.720	0.043	0.637 to 0.804	<0.001
	Borden’s method	0.723	0.048	0.629 to 0.816	<0.001
Kyphosis	C2-7 Cobb angle	0.871	0.042	0.789 to 0.954	<0.001
	C2-7 ARA	0.877	0.040	0.799 to 0.955	<0.001
	Borden’s method	0.765	0.068	0.632 to 0.897	<0.001
SVA > 40 mm	C2-7 SVA	0.913	0.044	0.769 to 1.000	0.001

Abbreviations: AUC, area under the curve; SE, standard error; CI, confidence interval; ARA, absolute rotational angle; SVA, sagittal vertical axis.

**Table 3 ijerph-18-13033-t003:** Comparative diagnostic accuracy of the four measurements.

Alignment	Measurements	Cut-Off	Sensitivity	Specificity	PPV	NPV	Accuracy
	C2-7 Cobb angle (°)	−8.5	0.676	0.704	0.696	0.685	0.690
Lordosis	C2-7 ARA (°)	−12.5	0.512	0.823	0.788	0.567	0.648
	Borden’s method (mm)	3.5	0.632	0.712	0.444	0.841	0.690
	C2-7 Cobb angle (°)	−4.5	0.905	0.711	0.352	0.977	0.739
Kyphosis	C2-7 ARA (°)	0.5	0.765	0.888	0.481	0.965	0.873
	Borden’s method (mm)	−1.5	0.611	0.863	0.393	0.939	0.831
SVA > 40 mm	C2-7 SVA (mm)	19.5	0.800	0.956	0.400	0.992	0.951

Abbreviations: PPV, positive predictive value; NPV, negative predictive value; ARA, absolute rotational angle; SVA, sagittal vertical axis.

## Data Availability

The data presented in this study are available on request from the corresponding author.

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
