# Peer review of "Diagnostic Accuracy of Magnetic Resonance Imaging for Sagittal Cervical Spine Alignment: A Retrospective Cohort Study"

_ijerph, 2021, doi:10.3390/ijerph182413033_

Round 1

Reviewer 1 Report

Manuscript with the results of an interesting and well thought-out retrospective study.

The problem is in the proposed study technique (MRI) which certainly has the advantage of not using ionizing radiation, but also of not allowing the patient to be studied in an upright position,

Difficult to compare data obtained from techniques that study patients differently (ortho and supine position).

therefore, as admitted by the authors themselves, with limits of comparison with conventional radiology.

Author Response

Reviewer #1

The problem is in the proposed study technique (MRI) which certainly has the advantage of not using ionizing radiation, but also of not allowing the patient to be studied in an upright position,

Difficult to compare data obtained from techniques that study patients differently (ortho and supine position).

therefore, as admitted by the authors themselves, with limits of comparison with conventional radiology.

Response: Thank you for the comment. Yes, as you commented, it is difficult to compare data obtained from different modalities. We considered the result could be clinically meaningful when there is limited information regarding the alignment (e.g. no upright image available) due to physical or medical constraints. 

Reviewer 2 Report

The manuscript “Diagnostic accuracy of magnetic resonance imaging for sagittal cervical spine alignment: a retrospective cohort study” by Chahyun Oh, Chan Noh, Jieun Lee, Sangmin Lee, Boohwi Hong, Youngkwon Ko, Chaeseong Lim, Sun Yeul Lee, Yoon-Hee Kim is an article that aimed to assessed the diagnostic accuracy of MRI and upright cervical radiography in the assessment of cervical spine alignment by using four well-known measurements.

Below are my comments and remarks regarding the article:

1. The introduction does not correspond to the issues raised - describing the factors influencing, for example, smartphones, cervical lordosis is not the subject of research or comparisons between research groups. By the way, it would be an interesting comparative study.
2. The description of the results could be wider.
3. In the discussion, the influence of factors on cervical alignment was discussed, although the results of the study only show a correlation between X-ray and MRI without taking into account the age, gender, BMI or any factors. These results should be compared with other authors and their correlations and comparisons between the groups should be added.

Author Response

Reviewer #2

The manuscript “Diagnostic accuracy of magnetic resonance imaging for sagittal cervical spine alignment: a retrospective cohort study” by Chahyun Oh, Chan Noh, Jieun Lee, Sangmin Lee, Boohwi Hong, Youngkwon Ko, Chaeseong Lim, Sun Yeul Lee, Yoon-Hee Kim is an article that aimed to assessed the diagnostic accuracy of MRI and upright cervical radiography in the assessment of cervical spine alignment by using four well-known measurements.

Below are my comments and remarks regarding the article:

  1. The introduction does not correspond to the issues raised - describing the factors influencing, for example, smartphones, cervical lordosis is not the subject of research or comparisons between research groups. By the way, it would be an interesting comparative study.

Response: Thank you for the comment. We reviewed the introduction and found that the paragraph describing smart phone use is somewhat distracting and decided to remove it from the introduction. And a paragraph in the introduction was revised as follows:

“Although radiography in an upright position is considered the standard method for assessing sagittal cervical alignment, cervical alignment is frequently determined or commented based on magnetic resonance imaging (MRI) or computed tomography (CT) performed in a supine position. However, the diagnostic cut-off values and accuracy of these determinations remain unclear. Also, standard upright imaging is not always available due to medical or physical constraints. The present study, therefore, retrospectively assessed the diagnostic accuracy of MRI using upright cervical radiography as a reference in the assessment of cervical spine alignment by using four well-known measurements.”

We considered that the issue of the increasing use of handheld device and the accompanying increasing trend of cervical malalignment is worth discussion. Although this issue does not directly relate to the main issue of the study (accuracy of cervical MRI for the assessment of cervical alignment), it raises importance of the assessment of cervical alignment itself. Thus, this issue was described in the discussion as follows:

 “Recently, a large-scale retrospective cross-sectional study* revealed that there was an increasing trend of loss of lordotic curvature across time during 2006 to 2018, especially in younger patients. The authors of the study pointed out that predominant use of internet and handheld devices may associated with this increase of cervical malalignment in younger patients. Many handheld device users flex their neck while viewing the device, which can cause harm to the cervical spine. As this concerning prevalence of abnormal curvature is expected to continue, the clinical importance of the assessment of cervical alignment is growing.”

* Shin, Y.; Han, K.; Lee, Y.H. Temporal Trends in Cervical Spine Curvature of South Korean Adults Assessed by Deep Learning System Segmentation, 2006-2018. JAMA Netw Open 2020, 3, e2020961, doi:10.1001/jamanetworkopen.2020.20961.

  1. The description of the results could be wider.

Thank you for the comment. As stratification of our results by sex could be a clinical interest, the Table 1 was revised accordingly. Also, the difference between sex was shown in Figure 3 (scatter plots of radiographic and MRI measurements). In response to your comment #3, we added an exploratory analysis to explore influence of age on the cervical alignment. Unfortunately, however, the other factors (height, weight, and BMI) were not able to be included in the analysis due to missing values. Followings were added in the results:

“The characteristics of these patients stratified by sex are summarized in Table 1. Height was not recorded in 72 patients and weight was not recorded in 67. C2-7 SVA on radiography was significantly lower in female patients. Other parameters showed no significant difference by sex.”

“The result of the exploratory analysis showed no significant correlation between age and the four measurements (Supplementary Fig. 2).”

  1. In the discussion, the influence of factors on cervical alignment was discussed, although the results of the study only show a correlation between X-ray and MRI without taking into account the age, gender, BMI or any factors. These results should be compared with other authors and their correlations and comparisons between the groups should be added.

Thank you for the invaluable comment. As noted above, such factors (age and gender) were considered in the revised result. And regarding this issue, following paragraphs were added in the discussion:

“The results of the exploratory analysis in this study showed that no significant correlation between age and cervical curvature exists. Nonetheless, this may need careful interpretation. In contrast to the current study, a previous study[1] showed clear correlation between age and lordosis (increased lordosis with increased age). This may be due to the differences between the cohort involved in these studies. In the previous study, most of the patients were females (70.4%), and relatively younger (mean age 48.1) with evenly distributed age. On the contrary, the cohort in the current study had a similar number of male and female patients with unevenly distributed age (tilted toward older ages). There were only less than 23 (20%) patients that were at ages under 40 in this study. Therefore, the correlation between age and cervical curvature warrants further study.”

“Third, since there were many missing data on patients’ weight and height, detailed analysis regarding the influence of these factors on the cervical alignment was not available.”

Also, several re-arrangements of the paragraphs were done in the discussion.  

Round 2

Reviewer 1 Report

The problem of limits of comparison with conventional radiology  is not solved.

Author Response

Thank you for the comment. The issue you pointed out is directly addressing the core theme of this study. As the two imaging modalities were taken in two different positions (i.e. upright for radiography and supine for MRI), direct comparison may not be possible. Therefore, the initial motivation of this study was not to insist that it is possible to replace radiography with MRI for the assessment of cervical alignment, but rather it was to assess validity of the unverified interpretations which were done in precedent studies. For example, Chen et al. (doi:10.1148/radiol.2213010365) had assessed cervical alignments of 64 cervical spondylotic myelopathy patients using MRI. As their determinations did not consider the positional change of cervical alignment, it may be misleading. Indeed, it turns out that according to the result of our study, the cut-off value of kyphosis determined by Borden’s method was decreased from 0 mm in upright radiography to -1.5 mm in MRI. So, we totally agree with your point that direct comparison between different imaging modalities is limited. We revised our introduction, discussion, and conclusion as follows in order to state this point more clearly:

The introduction was revised as follows:

“Although radiography in an upright position is considered the standard method for assessing sagittal cervical alignment[5], cervical alignment is frequently determined or commented based on magnetic resonance imaging (MRI) or computed tomography (CT) performed in a supine position[6-8]. However, difference in the patient positioning and the use of head stabilizer during MRI scan may affect the alignment. Therefore, direct comparison between these two modalities may be misleading. Also, the diagnostic cut-off values and accuracy of these determinations remain unclear. The present study, therefore, retrospectively assessed the diagnostic accuracy of MRI in the assessment of cervical spine alignment by comparing the MRI images with upright cervical radiography as a reference using four well-known measurements.” 

The discussion was revised as follows:

“This study showed that MRI measurements of cervical alignment in the supine position correlated significantly with upright radiographic measurements. Although the relationship between these two imaging modalities can allow cervical alignment to be predicted based on MRI measurements, these relationships are neither perfectly linear nor do they have regression coefficients near 1.0. Thus, MRI measurements cannot completely replace radiographic measurements for the assessment of cervical alignment, nor can the same cut-off values be applied.

Based on the result of the current study, kyphosis and C2-7 SVA > 40 mm can be reliably ruled out by the measurements on MRI. The evaluations of their negative predictive values showed excellent accuracy for all four methods. Lack of kyphosis on MRI was highly indicative of lack of kyphosis on radiography. In contrast, ‘loss of lordosis’ on MRI may not indicate a similar finding on radiography. The likelihood that ‘loss of lordosis’ is correct based on the MRI measurement using C2-7 ARA was only 56.7%. If ‘loss of lordosis’ is observed on MRI, it can be a result from kyphotic change due to positional change from upright to supine. Therefore, determination of cervical alignment based on MRI is limited. Also, distinct cut-off values as suggested in this study should be used to minimize the compromise of diagnostic accuracy.”

The conclusion was revised as follows:

“In conclusion, this study showed that MRI measurements of cervical alignment correlated significantly with upright radiographic measurements. MRI measurements may therefore be predictive of cervical alignment, especially for the exclusion of kyphosis and C2-7 SVA > 40mm. However, caution is needed in the other determinations using MRI, as their accuracies are limited.”

Note that a reference was added as a previous example study which used MRI for the assessment of cervical alignment (citation number 7)

Linsenmaier, U.; Deak, Z.; Krtakovska, A.; Ruschi, F.; Kammer, N.; Wirth, S.; Reiser, M.; Geyer, L. Emergency radiology: straightening of the cervical spine in MDCT after trauma--a sign of injury or normal variant? Br J Radiol 2016, 89, 20150996, doi:10.1259/bjr.20150996

Note that Table 1 was further revised to add more useful information and an additional figure (newly designated as Figure. 4) was added. The added figure shows correlation between the reference alignment (radiography) and the gap between the measurements of radiography and MRI. And following description was added in the result:

“There were significant correlations between the radiographic measurements and the difference between the two paired measurements (radiographic – MRI) (C2-7 Cobb angle: 0.405, C2-7 ARA: 0.404, Borden’s method: 0.533, C2-7 SVA: 0.837; p < 0.001 each) (Fig. 4). Patients with more lordotic curvatures on radiography showed greater kyphotic change on MRI. Also, patients with greater C2-7 SVA on radiography tended to show larger decrease of C2-7 SVA on MRI.”

Reviewer 2 Report

There is still a distraction between the aim of the study (retrospectively assessed the diagnostic accuracy of MRI using upright cervical radiography as a reference in the assessment) and the information presented in the introduction and discussion on the factors influencing lordosis and kyphosis of the spine.

Author Response

Reviewer #2

There is still a distraction between the aim of the study (retrospectively assessed the diagnostic accuracy of MRI using upright cervical radiography as a reference in the assessment) and the information presented in the introduction and discussion on the factors influencing lordosis and kyphosis of the spine.

Thanks for your invaluable comment. After careful consideration of your comment, we found that the description of the factors influencing cervical alignment is indeed distracting. Such statements were carefully subtracted from the manuscript. At the same time, we preserved statements addressing the growing importance of the assessment of cervical alignment as briefly as possible. We considered that introducing this contextual background in the introduction and the discussion is necessary.

The introduction was revised as follows:

“Cervical malalignment has been associated with the development of cervical pathologies[1,2] and reduced health-related quality of life[3]. A recent large-scale retrospective study[4] revealed that there is an increasing trend of cervical malalignment. Assessing cervical spine alignment, therefore, has become increasingly important.

Although radiography in an upright position is considered the standard method for assessing sagittal cervical alignment[5], cervical alignment is frequently determined or commented based on magnetic resonance imaging (MRI) or computed tomography (CT) performed in a supine position[6-8]. However, difference in the patient positioning and the use of head stabilizer during MRI scan may affect cervical alignment. Therefore, direct comparison between these two modalities may be misleading. Also, the diagnostic cut-off values and accuracy of these determinations remain unclear. The present study, therefore, retrospectively assessed the diagnostic accuracy of MRI in the assessment of cervical spine alignment by comparing the MRI images with upright cervical radiography as a reference using four well-known measurements[5,9].”

The discussion was revised as follows:

“This study showed that MRI measurements of cervical alignment in the supine position correlated significantly with upright radiographic measurements. Although the relationship between these two imaging modalities can allow cervical alignment to be predicted based on MRI measurements, these relationships are neither perfectly linear nor do they have regression coefficients near 1.0. Thus, MRI measurements cannot completely replace radiographic measurements for the assessment of cervical alignment, nor can the same cut-off values be applied.

Based on the result of the current study, kyphosis and C2-7 SVA > 40 mm can be reliably ruled out by the measurements on MRI. The evaluations of their negative predictive values showed excellent accuracy for all four methods. Lack of kyphosis on MRI was highly indicative of lack of kyphosis on radiography. In contrast, ‘loss of lordosis’ on MRI may not indicate a similar finding on radiography. The likelihood that ‘loss of lordosis’ is correct based on the MRI measurement using C2-7 ARA was only 56.7%. If ‘loss of lordosis’ is observed on MRI, it can be a result from kyphotic change due to positional change from upright to supine. Therefore, determination of cervical alignment based on MRI is limited. Also, distinct cut-off values as suggested in this study should be used to minimize the compromise of diagnostic accuracy.”

“Although symptoms, rather than anatomic changes in imaging results, are considered important, measured cervical alignment itself has clinical value in the pathophysiology and prognosis of the cervical spine. Cervical malalignment was shown to have negative effects on degenerative changes[2] and disc herniation[1] of the cervical spine. Also, cervical malalignment has been associated with increased neck disability and reduced health-related quality-of-life (HRQOL)[3,13,16]. Besides, according to a recent study[4], there was an increasing trend of loss of lordotic curvature across time during 2006 to 2018, especially in younger patients. The authors of the study pointed out that predominant use of internet and handheld devices may associated with this increase of cervical malalignment in younger patients. Many handheld device users flex their neck while viewing the device[17-20], which can cause harm to the cervical spine[21,22]. As this concerning prevalence of abnormal curvature is expected to continue, the clinical importance of the assessment of cervical alignment is growing.”

The conclusion was revised as follows:

“In conclusion, this study showed that MRI measurements of cervical alignment correlated significantly with upright radiographic measurements. MRI measurements may therefore be predictive of cervical alignment, especially for the exclusion of kyphosis and C2-7 SVA > 40mm. However, caution is needed in the other determinations using MRI, as their accuracies are limited.”

Note that a reference was added as a previous example study which used MRI for the assessment of cervical alignment (citation number 7)

Linsenmaier, U.; Deak, Z.; Krtakovska, A.; Ruschi, F.; Kammer, N.; Wirth, S.; Reiser, M.; Geyer, L. Emergency radiology: straightening of the cervical spine in MDCT after trauma--a sign of injury or normal variant? Br J Radiol 2016, 89, 20150996, doi:10.1259/bjr.20150996

Note that Table 1 was further revised to add more useful information and an additional figure (newly designated as Figure. 4) was added. The added figure shows correlation between the reference alignment (radiography) and the gap between the measurements of radiography and MRI. And following description was added in the result:

“There were significant correlations between the radiographic measurements and the difference between the two paired measurements (radiographic – MRI) (C2-7 Cobb angle: 0.405, C2-7 ARA: 0.404, Borden’s method: 0.533, C2-7 SVA: 0.837; p < 0.001 each) (Fig. 4). Patients with more lordotic curvatures on radiography showed greater kyphotic change on MRI. Also, patients with greater C2-7 SVA on radiography tended to show larger decrease of C2-7 SVA on MRI.”
